# Effects of potent neutralizing antibodies from convalescent plasma in patients hospitalized for severe SARS-CoV-2 infection

Arvind Gharbharan[1,16], Carlijn C. E. Jordans[1,16], Corine GeurtsvanKessel [1], Jan G. den Hollander[2], Faiz Karim[3], Femke P. N. Mollema[4], Janneke E. Stalenhoef – Schukken [5], Anthonius Dofferhoff[6], Inge Ludwig[7], Adrianus Koster[8], Robert-Jan Hassing[9], Jeannet C. Bos[10], Geert R. van Pottelberge[11], Imro N. Vlasveld[12], Heidi S. M. Ammerlaan[13], Elena M. van Leeuwen – Segarceanu[14], Jelle Miedema[1], Menno van der Eerden[1], Thijs J. Schrama [1], Grigorios Papageorgiou[1], Peter te Boekhorst[1], Francis H. Swaneveld[15], Yvonne M. Mueller [1], Marco W. J. Schreurs[1], Jeroen J. A. van Kampen[1], Barry Rockx [1], Nisreen M. A. Okba [1], Peter D. Katsikis [1], Marion P. G. Koopmans [1], Bart L. Haagmans [1], Casper Rokx [1,16✉] & Bart J. A. Rijnders[1,16]

In a randomized clinical trial of 86 hospitalized COVID-19 patients comparing standard care to treatment with 300mL convalescent plasma containing high titers of neutralizing SARS-CoV-2 antibodies, no overall clinical benefit was observed. Using a comprehensive translational approach, we unravel the virological and immunological responses following treatment to disentangle which COVID-19 patients may benefit and should be the focus of future studies. Convalescent plasma is safe, does not improve survival, has no effect on the disease course, nor does plasma enhance viral clearance in the respiratory tract, influence SARS-CoV-2 antibody development or serum proinflammatory cytokines levels. Here, we show that the vast majority of patients already had potent neutralizing SARS-CoV-2 antibodies at hospital admission and with comparable titers to carefully selected plasma donors. This resulted in the decision to terminate the trial prematurely. Treatment with convalescent plasma should be studied early in the disease course or at least preceding autologous humoral response development.

[1] Erasmus MC, University Medical Center, Rotterdam, The Netherlands. [2] Maasstad Hospital, Rotterdam, The Netherlands. [3] Groene Hart Hospital, Gouda, The Netherlands. [4] Haaglanden Medical Center, The Hague, The Netherlands. [5] OLVG Hospital, Amsterdam, The Netherlands. [6] Canisius Wilhelmina Hospital, Nijmegen, The Netherlands. [7] Bernhoven Hospital, Uden, The Netherlands. [8] Viecuri Medical Center, Venlo, The Netherlands. [9] Rijnstate Hospital, Arnhem, The Netherlands. [10] Reinier de Graaf Gasthuis, Delft, The Netherlands. [11] ZorgSaam Hospital, Terneuzen, The Netherlands. [12] Martini Hospital, Groningen, The Netherlands. [13] Catharina Hospital, Eindhoven, The Netherlands. [14] Sint Antonius Hospital, Nieuwegein, The Netherlands. [15] Unit of Transfusion Medicine, Sanquin Blood Supply, Amsterdam, The Netherlands. [16] These authors contributed equally: Arvind Gharbharan, Carlijn C.E. Jordans, Casper Rokx, Bart J.A. Rijnders. ✉email: c.rokx@erasmusmc.nl

Severe acute respiratory syndrome coronavirus 2 (SARS-CoV-2), the cause of coronavirus disease 2019 (COVID-19), continues to put a tremendous strain on healthcare systems despite the advances that were made regarding the management of these patients. Anti-inflammatory therapy with dexamethasone significantly decreased mortality[1]. Its beneficial effect is well documented after at least 7 days of symptoms and when patients need supplemental oxygen or admission to an intensive care unit (ICU). The role of direct antiviral therapy is less well-established, as a beneficial effect was observed in one but not in a second, much larger, randomized trial on remdesivir[2,3]. Furthermore, even in resource-rich countries, the drug has been out of stock repeatedly. All other repurposed antiviral drugs studied so far have failed to show any benefit. Clearly, there is an unmet need for antiviral therapy with well-established efficacy and global availability.

SARS-CoV-2-neutralizing antibodies are considered a promising treatment for COVID-19 and highly potent monoclonal antibodies are being studied[4-6]. Convalescent plasma (ConvP) can contain high levels of SARS-CoV-2-neutralizing antibodies and could therefore be regarded as an antiviral alternative for monoclonal antibodies to treat COVID-19. Neutralizing antibodies can cause a reduction of virus infectivity by binding to the surface of viral particles, which results in blocking one of the steps of the viral replication cycle, also known as virus neutralization[7]. Neutralizing SARS-CoV-2 antibodies recognize regions of the Spike protein, mainly the receptor-binding domain (RBD), and inhibit viral infectivity by several mechanisms. The most important one is blocking the RBD–ACE-2 receptor interaction and such, preventing the attachment of SARS-CoV-2 to the epithelial cell surface[8,9]. ConvP is a potentially scalable option during viral outbreaks. During previous human outbreaks of SARS-CoV and Middle East respiratory syndrome coronavirus, ConvP was used as a therapy with some success according to several small studies[10-13]. In SARS-CoV-2, preclinical research indicates a protective effect of human ConvP containing high levels of neutralizing antibodies when administered to hamsters prior to SARS-CoV-2 infection[14].

Conclusive evidence for the effectiveness of ConvP as a treatment for human SARS-CoV-2 infection is, however, yet to be generated[15-20]. ConvP administration late in the disease at 30 days post onset of symptoms did not benefit severely ill COVID-19 patients from China[18]. Results from meta-analyses including also non-randomized observational cohorts suggested that ConvP may benefit only subsets of patients[21-25]. According to the NIH COVID-19 Treatment Guidelines panel statement, conclusive evidence in support of ConvP therapy in patients hospitalized for COVID-19 are lacking[26], despite its emergency use authorization by the U.S. Food and Drug Administration on the 23 August 2020[27].

The goal of this study was to evaluate in a randomized trial the efficacy of ConvP treatment in hospitalized COVID-19 patients. We hypothesized that the administration of ConvP with high titers of neutralizing antibodies would provide benefit to COVID-19 patients in terms of clinical symptoms, SARS-CoV-2 shedding, and normalization of inflammatory markers. Our study demonstrates, however, that ConvP treatment fails to provide benefit in general. We were able to substantiate that autologous neutralizing antibodies already present at hospital admission may explain this finding. Our data provide guidance for future trials of antibody-based therapy for COVID-19 as well as clinicians involved in COVID-19 patient care.

## Table 1 Baseline characteristics of COVID-19 patients.

| | SoC ($n = 43$) | ConvP ($n = 43$) |
|---|---|---|
| Male sex, n (%) | 33 (77) | 29 (67) |
| Age (years), median (IQR) | 63 (55–77) | 61 (56–70) |
| Duration of symptoms at inclusion (days), median (IQR) | 11 (6–16) | 9 (7–13) |
| Number of comorbidities, n (%) | | |
| Diabetes mellitus | 8 (19) | 13 (30) |
| Hypertension | 11 (26) | 11 (26) |
| Cardiac | 11 (26) | 9 (21) |
| Pulmonary | 11 (26) | 12 (28) |
| Cancer | 3 (7) | 5 (12) |
| Immunodeficiency | 6 (14) | 5 (12) |
| Chronic kidney disease | 6 (14) | 1 (2) |
| Liver cirrhosis | 0 | 1 (2) |
| CRP (mg/L), median (IQR) | 109 (705 (12)165) | 84 (50–133) |
| Ferritin (μg/L), median (IQR) | 709 (525–1311) | 702 (406–1060) |
| LDH (U/L), median (IQR) | 356 (291–507) | 336 (259–454) |
| Lymphocytes (×10⁹/L), median (IQR) | 0.95 (0.80–1.30) | 1.20 (0.80–1.53) |
| Bilirubin (μmol/L), median (IQR) | 8 (6–12) | 9 (5–13) |
| WHO COVID-19 disease severity score[a], n (%) | | |
| ≤2 | 0 | 0 |
| 3 | 1 (2) | 7 (16) |
| 4–5 | 34 (79) | 31 (72) |
| 6–7 | 8 (19) | 5 (12) |

[a]WHO 8-point COVID-19 disease severity score (at study inclusion for patients and highest score ever during disease course for donors) in which "0" is no clinical or virological evidence of infection; "1" is no limitation of activities; "2" is limitation of activities; "3" is hospitalized, no oxygen; "4" is oxygen by mask or nasal prongs; "5" is non-invasive ventilation or high-flow oxygen; "0" is intubation and mechanical ventilation; "7" is ventilation and additional organ support (vasopressors, renal replacement therapy, ECMO), and "8" is death.

## Results

**Demographic characteristics of COVID-19 patients**. The study enrollment period was from 8 April to 14 June 2020. During that period, a total of 204 patients from 14 Dutch hospitals with a reverse-transcriptase PCR (RT-PCR) confirmed SARS-CoV-2 infection and admitted for a moderate, severe, or life-threatening COVID-19 infection were screened for eligibility. All had symptomatic COVID-19 disease as evaluated by study physicians according to the guidelines set by the Dutch National Institute for Public Health and Environment[28]. The most common reason for patients to decline participation was fear of adverse events (Supplementary Fig. 1). A total of 86 COVID-19 patients were enrolled and randomized to standard of care (SoC; $n = 43$) or treatment with ConvP ($n = 43$) (Table 1).

Overall, 72% of the patients were male and median age was 63 years (interquartile range (IQR) 56–74). At inclusion, they had COVID-19-related symptoms for a median of 10 days (IQR 6–15) and had been admitted to the hospital for 2 days (IQR 1–3) in line with what was previously reported as median duration of symptoms at hospitalization[29,30]. A total of 13 patients were admitted to the ICU and mechanically ventilated, all for no longer than 96 h at inclusion. Patients randomized to SoC experienced symptoms for a median of 2 more days at inclusion and more people had World Health Organization (WHO) disease severity scores ≥ 4 (98% vs. 84%) or ≥6 (ICU admission (19% vs. 12%). Four out of five blood biomarkers associated with an unfavorable COVID-19 disease course (C-reactive protein (CRP), ferritin, lactate dehydrogenase (LDH), and lymphocyte count) were slightly more disadvantageous in the SoC group. The total number of comorbidities were 56 and 57, respectively. From 66 of the patients, blood samples for additional immunological evaluations could be retrieved at baseline. Of these, 34 were in

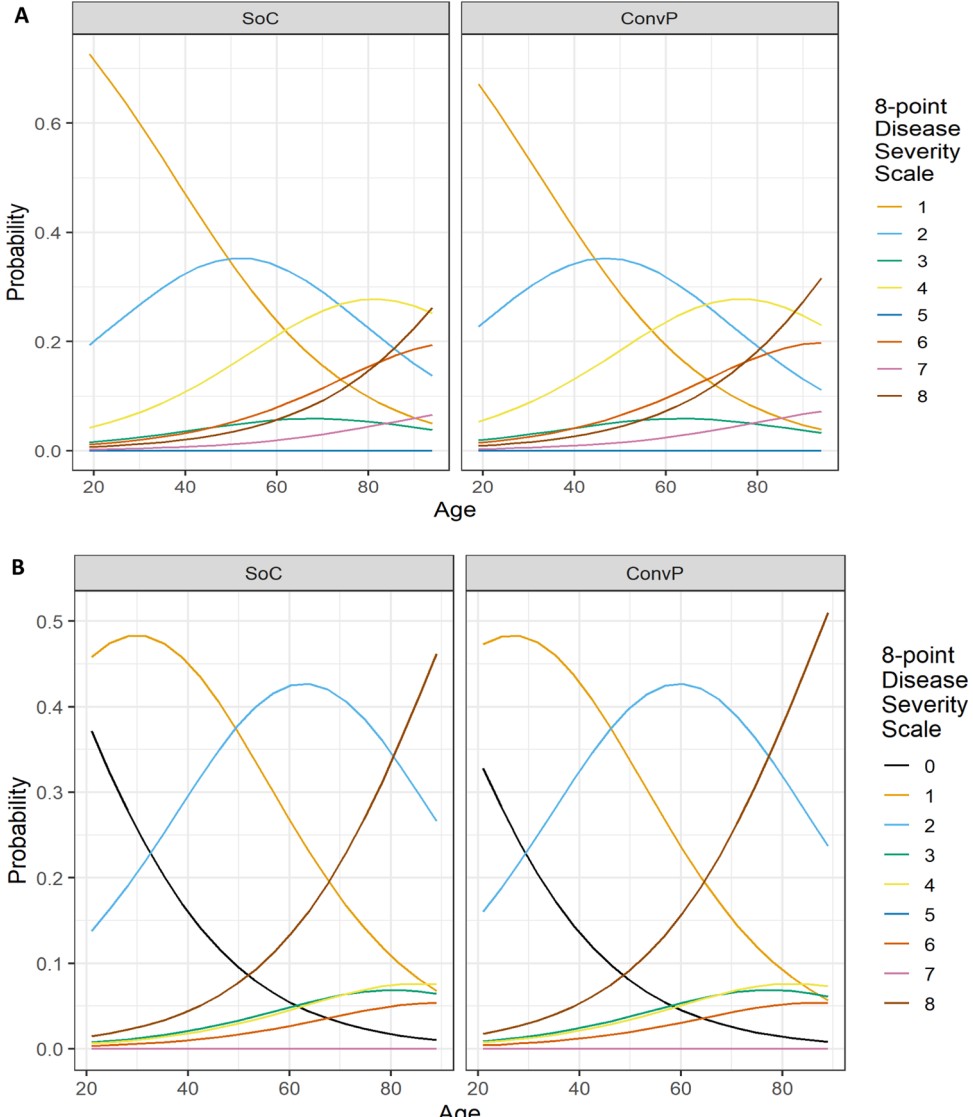

**Fig. 1 Predicted probabilities of WHO COVID-19 disease severity score. A** Disease severity score after 15 days. **B** Disease severity score after 30 days. Predicted probabilities of belonging to each outcome category of the WHO COVID-19 8-point Disease Severity Scale over different values of age and per treatment group. The predicted probabilities are based on the proportional ordinal logistic regression model adjusted for age, sex, CRP, and whether a patient was admitted to the intensive care unit at enrollment. The values of these factors are set to the median value (if continuous) or to the most frequent value (if categorical). WHO COVID-19 8-point Disease Severity Scale: 0 = no clinical or virological evidence of infection; 1 = no limitation of activities; 2 = limitation of activities; 3 = hospitalized, no oxygen therapy; 4 = oxygen by mask or nasal prongs; 5 = non-invasive ventilation or high-flow oxygen; 6 = intubation and mechanical ventilation; 7 = ventilation + additional organ support = vasopressors, renal replacing therapy, extracorporeal membrane oxygenation (ECMO); 8 = death.

the ConvP arm and their baseline characteristics were balanced with the overall group (Supplementary Table 1).

**Clinical outcomes.** Of the 43 patients randomized to ConvP, 6 (14%) died, whereas 11 of the 43 (26%) SoC patients died leading to an unadjusted odds ratio (OR) of 0.47 (95% confidence interval (95% CI): 0.15–1.38) for death. Although numerically higher, the predefined primary endpoint in the study protocol was an adjusted analysis of the overall mortality at day 60 after enrollment for patients treated with ConvP of which the OR was 0.95 (95% CI: 0.20–4.67, $p = 0.95$) (Supplementary Table 2). This adjusted analysis of mortality accounts for consistently reported predictors of death in COVID-19 patients[31–35]. By adjusting for the same confounders in a proportional odds ordinal logistic regression model, we identified age-specific probabilities for being

scored in specific categories of the eight-point WHO COVID-19 disease severity score at day 15 (Fig. 1A) and day 30 (Fig. 1B) after randomization. Notably, these probabilities were comparable between both study arms for all scores at day 15 ($p = 0.58$) and day 30 ($p = 0.67$) throughout the study (Supplementary Tables 3 and 4). An identical number of 25 (58%) patients in the ConvP and 25 (58%) in the SoC group had improved by day 15 on the WHO COVID-19 disease severity score (adjusted OR 1.30, 95% CI: 0.52–3.32, $p = 0.58$). Treatment with ConvP was also not associated with earlier discharge in Cox regression analysis (adjusted hazard ratio 0.88, 95% CI: 0.49–1.60, $p = 0.68$), also when cumulative incidences were corrected for the competing risk of death (Fig. 2 and Supplementary Fig. 2 and Supplementary Table 5). The median duration of admission was 8 days (IQR 3–21) and 8 days (IQR 4–19) for ConvP vs. SoC. Patients who

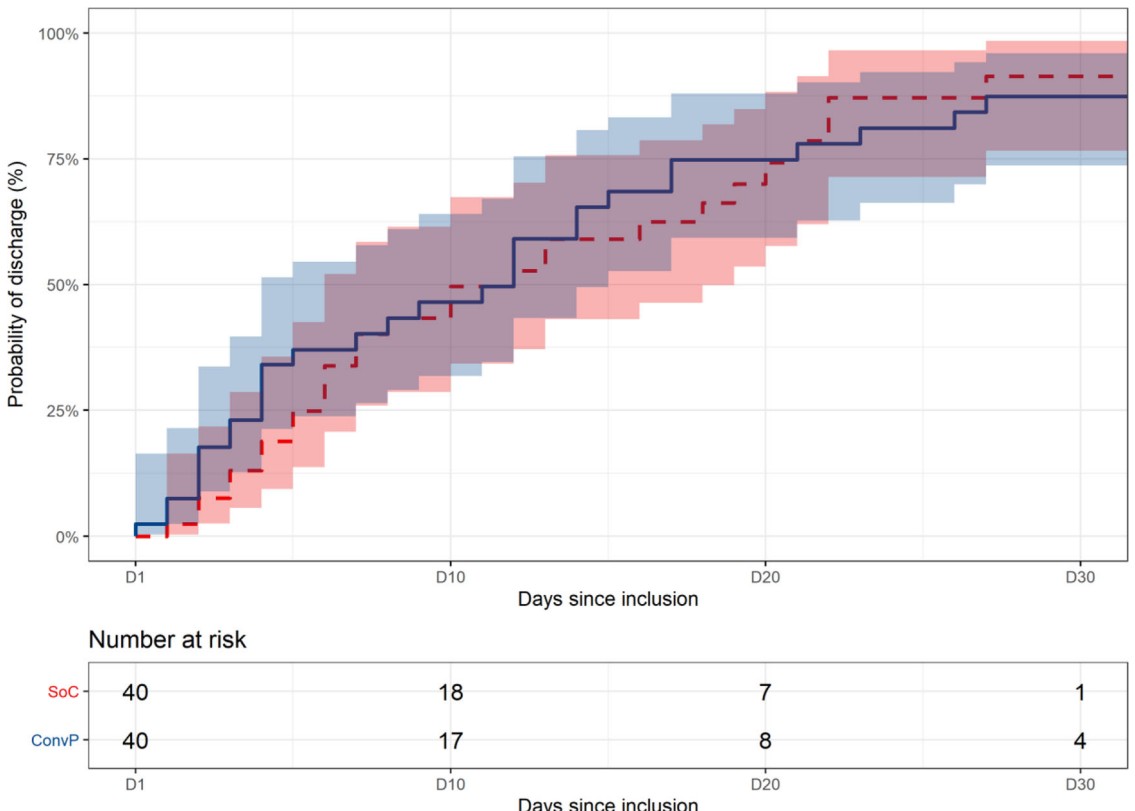

**Fig. 2 Kaplan–Meier curves of probability of discharge.** Shaded area indicates 95% CI around the Kaplan–Meier estimate of discharge for the two treatment groups (standard of care: red dashed line; convalescent plasma: blue solid line) after enrollment ($D = 1$) and table with number of subjects at risk of discharge. Death is not accounted for as competing risk.

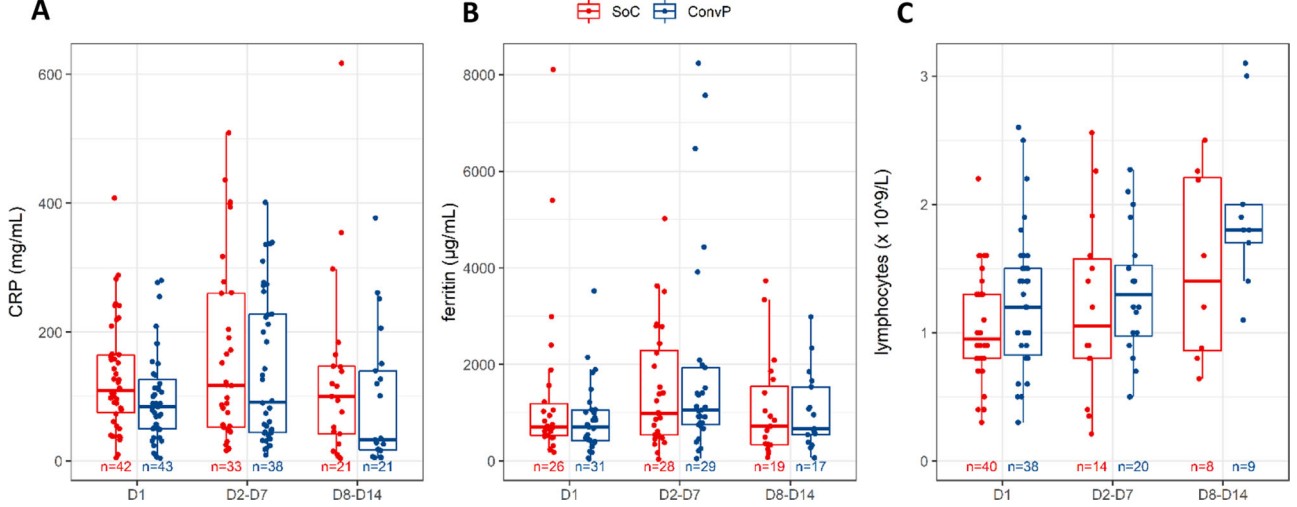

**Fig. 3 Inflammatory markers in patients. A** CRP. **B** Ferritin. **C** Lymphocytes. CRP, ferritin and lymphocytes were measured* in the serum of COVID-19 patients (standard of care: red; convalescent plasma: blue) on day 1 of enrollment, between days 2–7 and days 8–14 after enrollment. Reported data are the highest value for CRP and ferritin, and the lowest value for lymphocytes. Box plots indicate median (middle line), 25th, 75th percentile (box), and 5th and 95th percentile (whiskers), as well as outliers (single points). *Only measured if it was part of routine care.

died were 69 years (IQR 63–84) and 12 (71%) were men. Baseline data of patients who survived or died in each of the study arms are available in the Supplementary Table 6. No serious adverse events possible related to ConvP were observed.

Overall, no difference was observed between the highest measurements of CRP or ferritin (the 2 biomarkers included in the electronic case report form) 7 and 14 days after enrollment between groups (Fig. 3 and Supplementary Table 7). In a subset of

34 patients with data available, we noticed that the median absolute lymphocyte counts in the peripheral blood at these time points were comparable and followed similar trends in recovery in each treatment group within 2 weeks after enrollment.

**Donor characteristics.** Collectively, 3200 recovered COVID-19 donors volunteered in April 2020 as ConvP donors. The first 115 patients were selected, who had RT-PCR-proven COVID-19

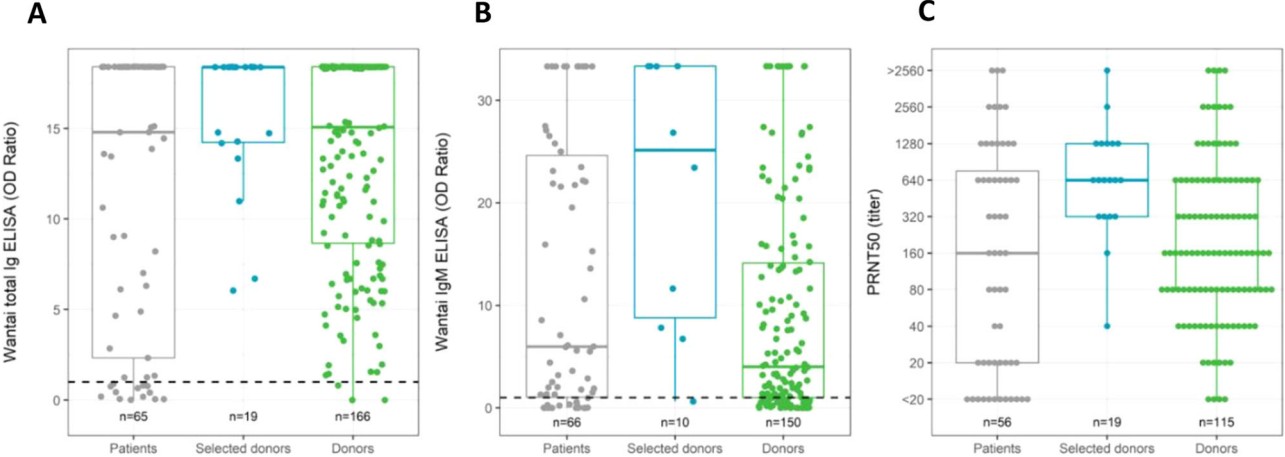

**Fig. 4 Antibodies against receptor-binding domain and viral neutralization capacity in patients and donors. A** SARS-CoV-2 total Ig against SARS-CoV-2 receptor-binding domain (RBD) measured by Wantai ELISA. **B** SARS-CoV-2 IgM against SARS-CoV-2 RBD measured by Wantai ELISA. **C** Viral neutralization capacity measured as PRNT50 titer. SARS-CoV-2 total Ig and IgM against SARS-CoV-2 RBD and viral neutralization capacity were evaluated in the serum of COVID-19 patients (gray) at enrollment (day 1) and serum of donors at day of plasma donation (all 115 donors tested: green; donors of whom plasma was selected for use in the study: blue). Box plots indicate median (middle line), 25th, 75th percentile (box), and 5th and 95th percentile (whiskers), as well as outliers (single points). Dashed line indicates positive cutoff at 1.0 optical density (OD) ratio for both total Ig and IgM.

disease, fulfilled the other inclusion criteria for ConvP collection, had a determination of SARS-CoV-2-neutralizing antibody responses (Supplementary Table 8) and completed the online questionnaire regarding donor characteristics. One hundred and five of the 115 donors were male, the median age was 43 years, and they had been symptomatic for a median of 12 days (IQR 8–18). Their disease course had been generally mild reflected by a 12% admission rate for COVID-19. Overall, we detected virus-specific total Ig and IgM antibodies against SARS-CoV-2 RBD by enzyme-linked immunosorbent assay (ELISA) in serum samples of 114 of 115 (99%) donors at median 34 days in their convalescent phase. The median total Ig and IgM optical density (OD) ratios in all donors were respectively 15.08 (IQR 8.60–18.41) and 4.03 (IQR 0.96–14.33). Although OD ratios from ELISAs correlate with neutralization capacity against SARS-CoV-2, substantial outliers with lower than expected neutralizing antibody titers are often observed and the correlation plateaus at increasing antibody levels leading to a loss of discriminative capacity to detect plasma with very high neutralizing antibody titers[36]. Therefore, a plaque-reduction neutralization test (PRNT) using the whole SARS-CoV-2 virus was used for the selection of donors for the study. In reporting the PRNT50 titers, the diluting factors are given. In 110 of 115 donors (96%) tested, neutralizing antibodies could be detected. The median PRNT50 titer was 160 (IQR 80–640) with 78% and 43% having a PRNT50 of at least 80 or 320, respectively. PRNT50 titers of 80 and 320 were previously shown to predict a <5% and <1% chance of demonstrating replication-competent virus in the upper respiratory airway of COVID-19 patients[37]. A titer above 80 was defined as the minimum neutralizing capacity required for a donor to be eligible for ConvP donation. Of the 19 donors of whom ConvP was eventually used, all but 2 had a PRNT50 titer of at least 320. These 19 selected individuals all had mild disease without hospitalization and a more recent resolution of symptoms (20 days) than other donors.

**Immunological analyses in COVID-19 patients.** Serum was available from 66 subjects for PRNT50 and ELISA testing at inclusion. Logistical issues at the peak of the pandemic prevented the collection of serum from the remaining 20 patients. At inclusion, 80% tested positive with the total Ig SARS-CoV-2 RBD

antibody test with OD ratios at 14.80 (IQR 1.84–18.41), while IgM SARS-CoV-2 RBD antibodies were present in 77% with OD ratios at 5.98 (IQR 0.86–25.19) (Fig. 4). Interestingly, the 19 selected donors had levels of SARS-CoV-2 RBD total Ig (OD ratio 18.39, IQR 14.19–18.39) that were comparable ($p = 0.78$) with the baseline levels in hospitalized patients who had been symptomatic for a median of 10 days, while SARS-CoV-2 RBD IgM antibodies tended to be higher (OD ratio 25.13, IQR 7.55–33.32; $p = 0.02$). The OD ratio of the total Ig ($p = 0.71$) and IgM ($p = 0.83$) SARS-CoV-2 RBD antibody between patients in the ConvP and SoC group were however not meaningfully different (Fig. 5A, B).

We further confirmed the presence of SARS-CoV-2 antibodies in the serum of COVID-19 patients by measuring nucleocapsid (N-protein) IgM and IgG antibodies, and comparing COVID-19 patients with a group of ConvP donors. Fifty-one patients had nucleocapsid IgM and IgG antibodies with a median of 5.65 units (IQR 2.59–11.73) and 16.19 units (IQR 6.91–30.48), respectively (Fig. 5C, D). Donors ($n = 54$) had nucleocapsid IgM and IgG antibodies with a median of 3.62 units (IQR 2.20–6.28) and 20.25 units (IQR 13.74–30.69), respectively. Using a cutoff of 11 units, 14 patients were positive for IgM antibodies (27.5%) and 31 for IgG antibodies (60.8%), whereas from the donors 11 were positive for IgM antibodies (20.4%) and 45 for IgG antibodies (80.3%). None of the nine healthy controls had IgM or IgG antibodies against SARS-CoV-2 nucleocapsid above the cutoff.

Next, to explore the functionality of detected antibodies, we used viral neutralization tests with SARS-CoV-2 from the same serum samples of patients and ConvP donors in whom the presence of SARS-CoV-2 RBD-specific antibodies were determined. This was possible in 56 of 66 enrolled patients due to limited serum availability after antibody testing in 10 patients. To our surprise, in 44 (79%) patients, neutralizing antibodies at PRNT50 of ≥20 were detected at median 160 (IQR 20–1280) (Fig. 6). As expected, this titer correlated ($r^2 = 0.36$, $p = 0.07$) with the duration of symptoms (Supplementary Fig. 3) and was comparable to the median titer observed in all 115 donors tested ($p = 0.4$). The median PRNT50 of the 19 selected donors (640, IQR 320–1280) was however higher $p = 0.011$, with 90% having a PRNT50 titer ≥320 compared to 46% ($p = 0.001$) in patients. Only 2 of the 19 selected donors had titers <320

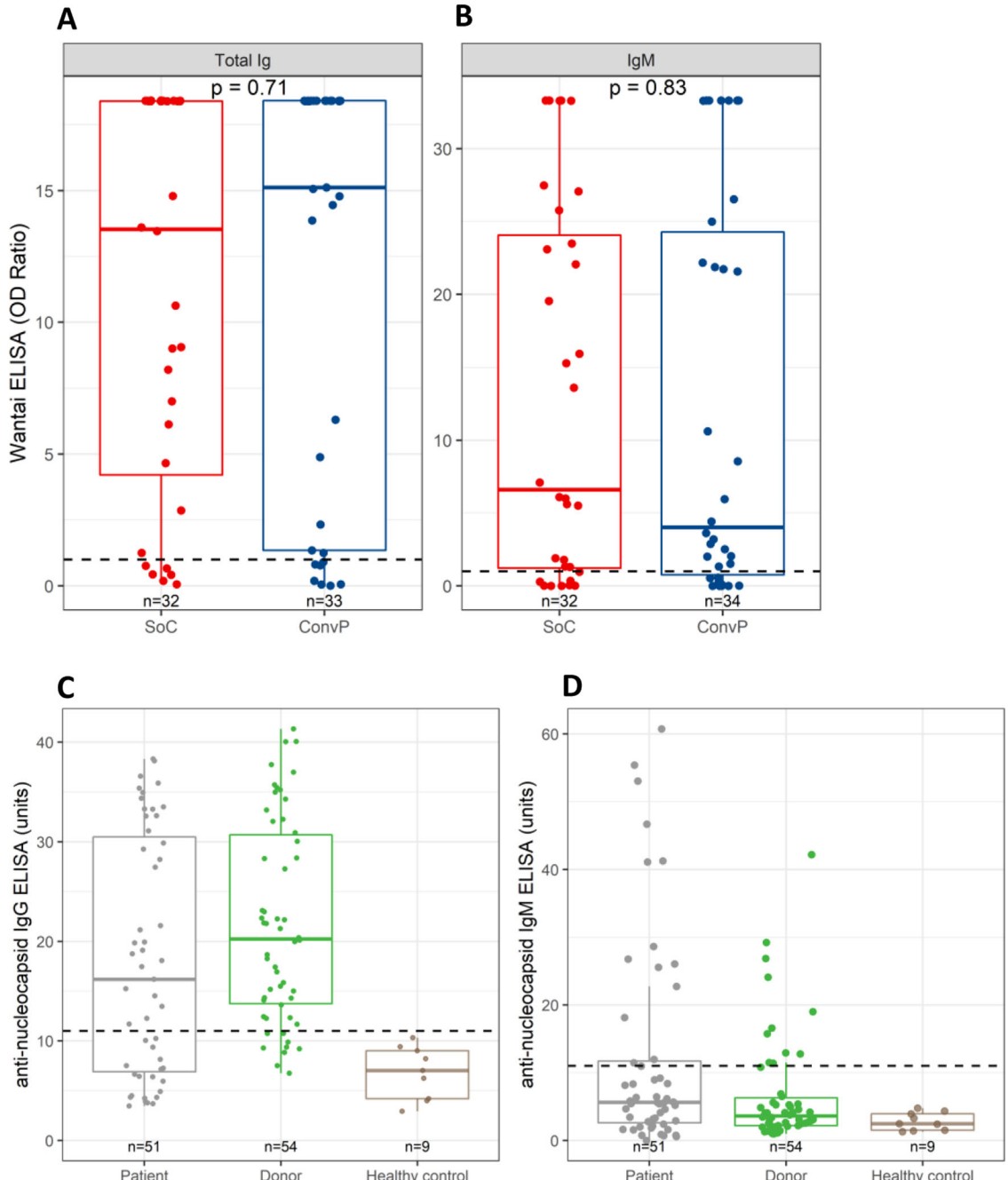

**Fig. 5 SARS-CoV-2 antibodies in patients, donors, and healthy controls. A** SARS-CoV-2 total Ig against SARS-CoV-2 receptor-binding domain (RBD) measured by Wantai ELISA. **B** SARS-CoV-2 IgM against SARS-CoV-2 RBD measured by Wantai ELISA. **C** Nucleocapsid IgG antibodies. **D** Nucleocapsid IgM antibodies. An independent Mann–Whitney *U*-test was used to evaluate SARS-CoV-2 total Ig and IgM against SARS-CoV-2 RBD measured by Wantai ELISA in the serum of COVID-19 patients (standard of care: red; convalescent plasma: blue) at enrollment (day 1). Nucleocapsid IgM and IgG antibodies were measured in the serum of COVID-19 patients (gray) at enrollment (day 1) in serum of donors (green) at week 6 post infection and in serum of healthy uninfected controls (brown). Box plots indicate median (middle line), 25th, 75th percentile (box), and 5th and 95th percentile (whiskers), as well as outliers (single points). Dashed line indicates positive cutoff at 1.0 OD ratio for both total Ig and IgM Wantai ELISA, and 11 units for both IgM and IgG nucleocapsid antibodies.

and these were administered to 2 patients with PRTN50 titers at >2560 and 640, respectively at baseline. These patients were admitted for 12 and 13 days, and both survived throughout day 60.

To independently confirm the high virus-neutralizing SARS-CoV-2 antibody levels in the included patients, we additionally tested serum from 37 RT-PCR-confirmed COVID-19 patients from the month preceding the start of the study from whom

serum samples from <72 h after hospital admission to a non-ICU ward were available at the Erasmus MC University Medical Center. With a median age of 65 years (IQR 56–74), 60% males, and symptom duration of 9 days (IQR 4–13), these patients were comparable to the study population as were their biomarkers that predict disease outcomes (Supplementary Table 9). We found that 26/37 (70%) of these patients had SARS-CoV-2 Ig antibodies including 23/37 (62%) at a ratio >10, indicating the presence

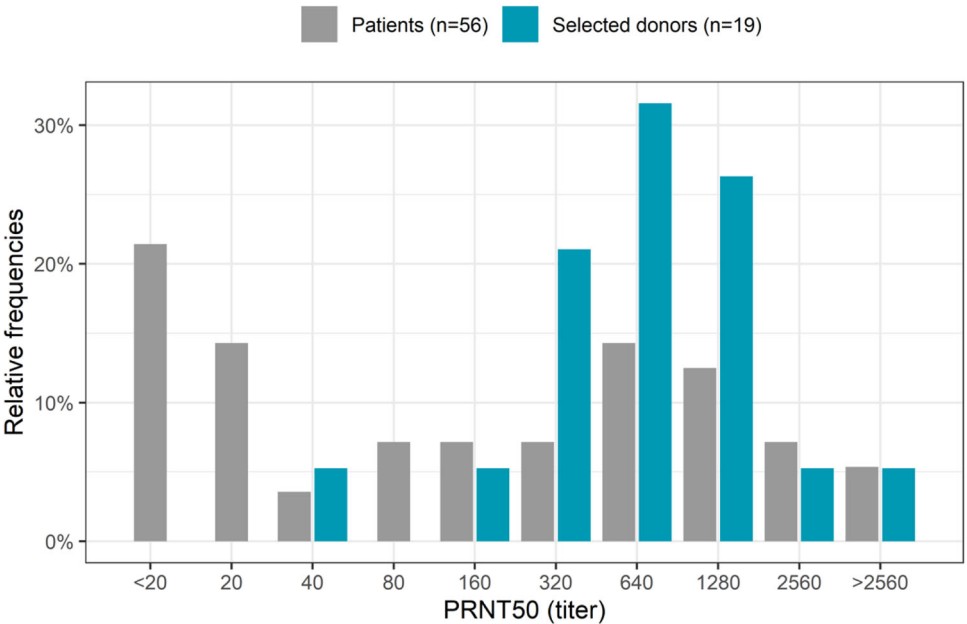

**Fig. 6 Relative frequencies of PRNT50 titers in patients and selected donors.** PRNT50 titers were measured in the serum of COVID-19 patients (gray) on day 1 of enrollment and in selected donors (blue). Height of the bars indicates the relative frequency of PRNT50 titer in that group.

of neutralization capacity based on our previous observation (Supplementary Fig. 4). [34]

Finally, to assess whether ConvP treatment had a more indirect effect on the COVID-19 disease course by potentially dampening the inflammatory response, we measured serum proinflammatory cytokines interleukin (IL)-6, tumor necrosis factor-α (TNFα), interferon-γ (IFNγ), IL-1β, IL-2, IL-4, IL-10, and IL-12p70. We investigated nine patients that received ConvP and ten SoC for which we had a complete set of serum samples for the first 2 weeks post inclusion in the study. For these patients, we compared the cytokine levels at enrollment (day 1), and at days 7 and day 14 after enrollment. There was no difference between the treatment arms on day 1 for IL-6, TNFα, and IFNγ. Importantly, the decrease in cytokine levels on day 7 and 14 were comparable between patients receiving ConvP and SoC, both for concentration (Fig. 7A) and fold change from day 1 (Fig. 7B). No differences were seen between groups for IL-1β, IL-2, IL-4, IL-10, and IL-12p70 (Supplementary Table 10). The above provides further evidence that plasma therapy had no effect on the inflammation and course of COVID-19.

**Virological analyses in COVID-19 patients.** Nasopharyngeal swabs were taken of 51, 54, 53, 15, and 45 patients at enrollment (day 1) and day 3, day 7, day 10, and day 14 after enrollment, respectively. In unadjusted analyses, the proportion of samples where SARS-CoV-2 genome was detectable by RT-PCR was higher in the SoC group at day 1 (82% vs. 67%), day 3 (79% vs. 46%), and day 14 (21% vs. 9%) compared to patients in the ConvP group. The calculated median SARS-CoV-2 viral loads were higher in the SoC group at inclusion ($5.7 \times 10^3$ copies/ml, IQR: $7.3 \times 10^2$–$4.7 \times 10^4$ vs. $1.4 \times 10^3$ copies/ml, IQR: 0–$1.7 \times 10^4$) and day 3 ($1.1 \times 10^3$ copies/ml, IQR: 0–$8.2 \times 10^3$ vs. 0 copies/ml, IQR: 0–$8.7 \times 10^1$). Thus, the apparent lower virus shedding in the upper airway was already present before ConvP initiation and this remained so over time in the treatment group without major appreciable influence of ConvP therapy (Fig. 8A). Indeed, after adjustment in a mixed model by covariates associated with COVID-19 disease severity as we had predefined in the study protocol (Supplementary Table 11), the slope of the viral load

decay from day 1 to 14 was estimated to be less steep in the ConvP group at −0.4 log copies/mL (95% CI −0.7 to −0.1) overall (Fig. 8B) and comparable when considering only patients with detectable SARS-CoV-2 in nasopharyngeal swabs at enrollment (Fig. 8C).

With regard to virus viability, we had 12 patients (6 in each arm, median 8 days of symptoms at inclusion) where we were able to do culture samples for SARS-CoV-2 replication, obtained after median 2 days following inclusion. Although a systematic collection of cultures before inclusion was not required per protocol and current knowledge indicates that viral cultures tend to become negative around 8 days of symptoms[37,38], none of the six patients from the SoC arm and one patient on ConvP had cytopathic effects (CPEs) in culture after 7 days of incubation, again signaling no added value of ConvP. This patient with a sample containing replication-competent virus had 6 days of symptoms at inclusion, without antibodies detected then, and died during follow-up.

## Discussion

Our study demonstrates that the administration of ConvP with high titers of virus-neutralizing antibodies does not benefit patients who are hospitalized for COVID-19 after 10 days post symptom onset. Overall, we found no improvement in key clinical, immunological, or virological parameters indicative of any effect favoring ConvP. This certainly does not exclude a possible beneficial effect in patients who have not yet started producing autologous neutralizing antibodies at the time of ConvP transfusion, but these patients were rare in our study population. When these data on the baseline antibody levels became available to us, we considered it highly unlikely that the current study design would allow for the detection of a significant clinical effect. After discussion with the Data Safety Monitoring Board (DSMB), the decision was made to interrupt study recruitment. Our data agree with the recent data from the Placid trial. In this study of 464 patients, half were randomized to ConvP. The intervention did not decrease the risk of progression to severe disease or death, nor did it decrease hospital stay. However, in contrast to our study, donors for the Placid trial were not screened for the

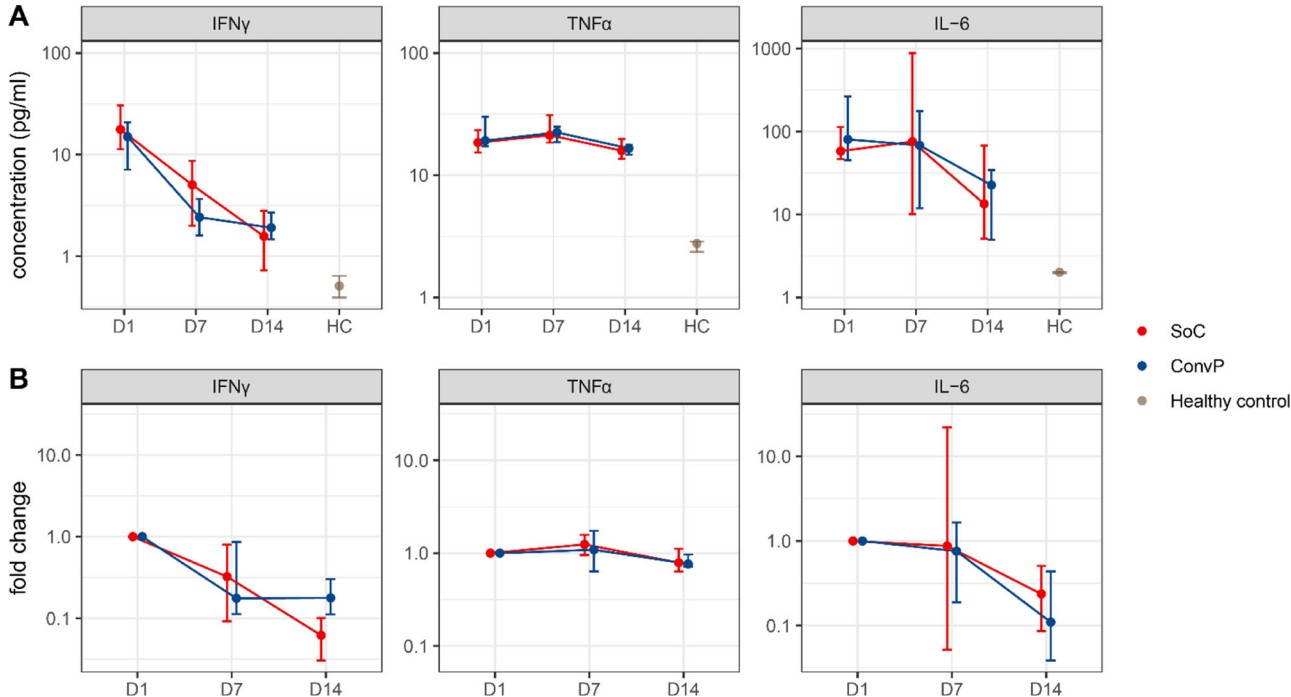

**Fig. 7 IFNγ, TNFα, and IL-6 in patients and healthy controls. A** Cytokine concentrations in serum. **B** Fold change. Cytokines IL-6, IFNγ, and TNFα were measured in the serum of nine convalescent plasma and ten standard of care patients (standard of care: red; convalescent plasma: blue) at enrollment (day 1) and on day 7 and day 14 after enrollment, and in the serum of healthy controls (brown). Dots represent median and vertical lines represent IQR.

presence of neutralizing antibodies before their plasma was used for the study. When this was done in retrospect, the PRNT50 titer in their donors turned out to more than 10-fold lower (40) than the median titer of 640 in our ConvP donors[39].

When the ConCOVID study was designed, the timing of neutralizing antibody development during SARS-CoV-2 infection was not well-established and certainly not common knowledge for those involved in COVID-19 care. We considered it unlikely that patients with severe disease requiring hospitalization would already have high titers of autologous neutralizing antibodies at the time of hospital admission. Although no formal stopping rule was reached when the study was discontinued, the study team and the DSMB members concluded that, considering the hypothesis of the study that was being tested, the chances of finding a significant difference in the primary endpoint even after full enrollment were too small to justify its continuation under its current design. Also, amending the study by excluding patients with autologous antibodies at screening was considered no option either, as it would leave too few eligible patients, as ~80% would have to be excluded.

Our observations are relevant for studies that continue to enroll hospitalized patients, as well as for emergency access, and compassionate use programs on ConvP for COVID-19. The data strongly suggest that any effect elicited by ConvP is more likely to occur when ConvP is given as early as possible in the disease course, which was also suggested by the cohort of the Mayo Clinic-led Expanded Access Program[17]. However, the latter was an observational study without a formal control arm and data on the time, as symptom onset were not reported. Using antibody-based therapy as early as possible after exposure to maximize their therapeutic effects is similar to the use of anti-hepatitis B virus or rabies immunoglobulin preparations[40,41]. Given the current data on the development of humoral anti-SARS-CoV-2 responses starting after ~1 week of symptoms, the window of opportunity is likely to be before day 7 after symptom onset and rapidly decreases thereafter[42,43]. Nevertheless, many ongoing

trials are now focusing on hospitalized patients and the time from disease onset to admission was repeatedly shown to be comparable to the 10 days in our study[2,30,44]. Therefore, in the vast majority of patients in studies with a comparable study design, the production of autologous humoral immunity against SARS-CoV-2 will have started as well. This is also supported by the notion that the generation and strength of the neutralizing antibody response correlates with disease severity[43,45,46]. We therefore predict that, on their own, almost all ongoing trials using a frequentist's design will be substantially underpowered to show beneficial effects from ConvP in hospitalized patients. This problem of insufficient statistical power may be partially circumvented by pooling data from ongoing trials together in real-time as suggested by others and as initiated under the COMPILE initiative in the United States[47].

Apart from correctly identifying patients who are likely to benefit most from ConvP, choosing the optimal ConvP donor with high titers of neutralizing antibodies is likely to be equally critical. Although data from formal dose-finding studies of ConvP are pending, the volume of and the minimum antibody titer in ConvP, but also the methods used to measure antibody titers, vary substantially across study protocols. The fact that this may turn out to be critical was demonstrated in a COVID-19 hamster model in which disease could be prevented with human ConvP with an exceptionally high neutralizing antibody titer of 2560, while ConvP with an above average antibody titer of 320 did not[14]. On theoretical grounds, ConvP will need to contain a certain (as of yet unknown) minimum level of neutralizing antibodies to ascertain an antiviral effect. Indeed, the antibodies in a standard 300 mL plasma unit will be approximately ten times diluted when given to an average human adult. With concurrent emerging data, including from our laboratory, it is likely that a neutralizing goal directed therapy to reach a minimum neutralization titer of 80 in vivo after transfusion should be obtained to recapitulate the 95% probability of inhibiting viral growth in vitro. This should then also take into account the

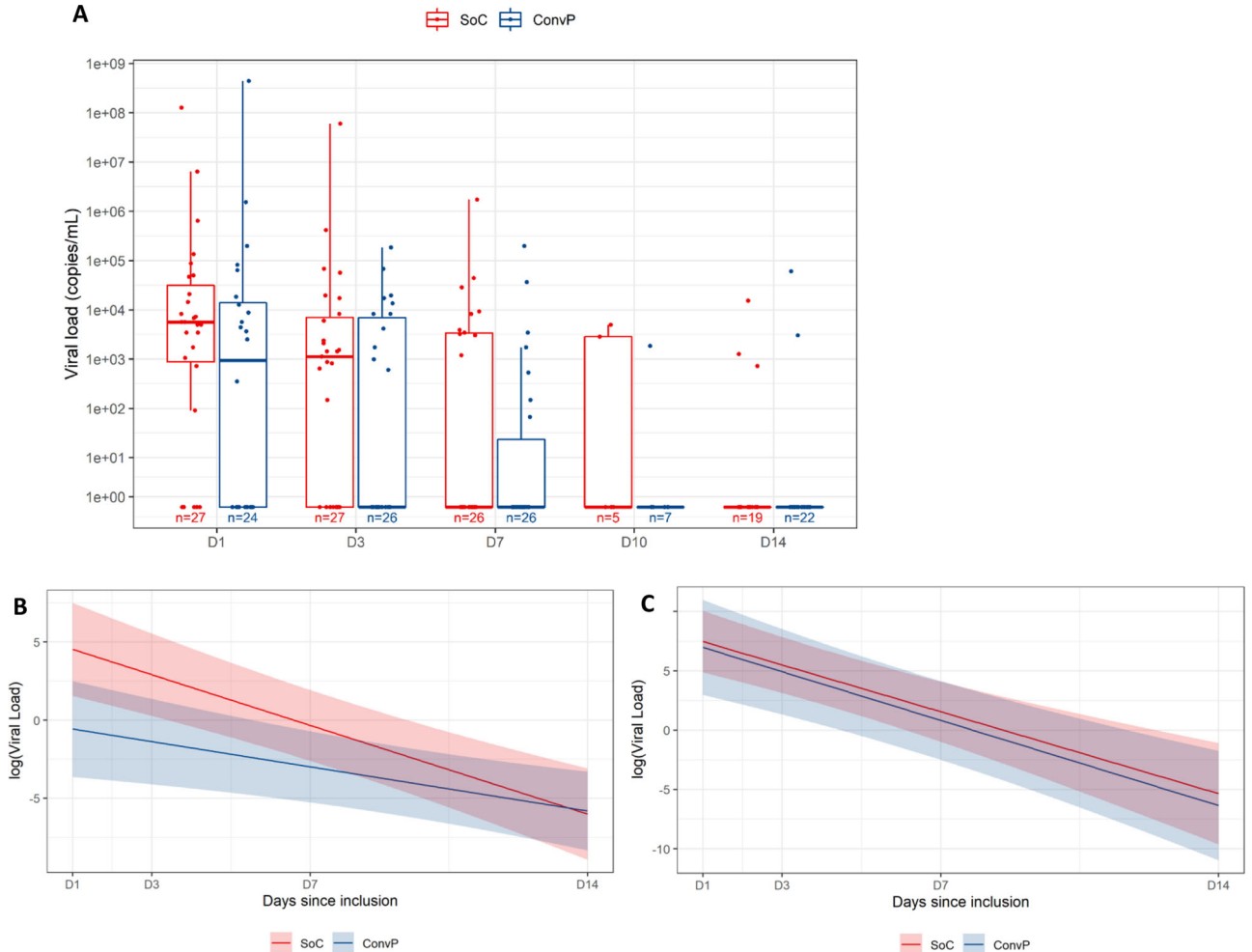

**Fig. 8 SARS-CoV-2 (predicted) viral load in patients. A** SARS-CoV-2 viral load in patients. **B** Predicted SARS-CoV-2 viral load in patients. **C** Predicted SARS-CoV-2 viral load in selected patients. SARS-CoV-2 viral load (copies/mL) measured by RT-PCR from nasopharyngeal swabs of COVID-19 patients (standard of care: red; convalescent plasma: blue) at enrollment (day 1) and day 3, day 7, day 10, and day 14 after enrollment. Box plots indicate median (middle line), 25th, 75th percentile (box), and 5th and 95th percentile (whiskers), as well as outliers (single points). Predicted evolution mean (solid line) and 95% CI (shaded area) of SARS-CoV-2 log(Viral Load) in log(copies/ml) per day since enrollment ($D = 1$) for COVID-19 patients (standard of care: red; convalescent plasma: blue). Predicted evolution mean (solid line) and 95% CI (shaded area) of absolute log(Viral Load) in log(copies/ml) per day since enrollment ($D = 1$) for COVID-19 patients (standard of care: red; convalescent plasma: blue) excluding subjects who had SARS-CoV-2 viral load equal to zero at day 1 of enrollment.

approximately tenfold dilution of ConvP in human plasma during transfusion. It is therefore worrisome if ConvP compassionate use programs and trials currently lack donor screening for neutralizing antibodies or select donors on ELISA testing only. Although data show that strong ELISA signals correlate with neutralization capacity, the predictive value of any ELISA cutoff for the presence of high levels of neutralizing antibody titers seems moderate at best[36]. Without readily available alternatives to ascertain antibody functionality, we consider the use of virus neutralization assays essential to avoid suboptimal donor selection. The use of hyperimmune Ig preparations produced from a large donor pool of ConvP (also called COVIg) and specific, highly neutralizing monoclonal antibodies may resolve this issue in the future[48,49].

Our study has several limitations. First, the premature ending prevents definite conclusions regarding the lack of clinical benefit of ConvP. The COMPILE real-time meta-analysis initiative described above should be able to solve this limitation[47]. Also, two large platform trials have opened a ConvP arm for hospitalized patients (the UK RECOVERY trial

and global REMAP-CAP)[50,51] and with their Bayesian design continue enrollment until futility or effectivity is documented. Second, the decision to end the study does not take into account that plasma could have effects unrelated to virus neutralization. We consider these effects highly unlikely, because far higher doses of plasma or immunoglobulins (usually 70–150 g) are used to reach these immunomodulatory effects than that present in a single unit of plasma (3 g). One concern with plasma treatment is whether antibody-dependent enhancement of infection could be mediated by the transferred antibodies. The findings described by Joyner et al.[17,52] on over 35,000 ConvP transfusions, including many with lower antibody titers than in our study, is reassuring in that perspective. Finally, we did not record the use of corticosteroids for COVID-19. During the recruitment for the trial, corticosteroids for non-ICU patients were not recommended in the Dutch COVID-19 guideline.

For future directions, our data support a more prominent role for ConvP early in the disease course, potentially in the outpatient setting, in particular in those with a higher risk of disease progression. It could also serve as a way to protect B-cell-depleted

patients or as post exposure prophylaxis after high-risk exposure[53]. Selecting hospitalized patients for ConvP treatment based on their antibody test results seems a logical way forward when plasma with high titers of neutralizing antibodies is scarce. Finally, studies in hospitalized patients will have to be sufficiently large to document a therapeutic benefit independent of dexamethasone and remdesivir therapy. In conclusion, no beneficial effects of ConvP were observed in patients recently hospitalized with COVID-19. The most likely explanation is the already high antibody titers on the day of inclusion. ConvP to treat COVID-19 should be targeted to patients as early as possible in their disease course and before a strong autologous neutralizing humoral response can be observed.

## Methods

**Study design and population**. The ConCOVID study was a multicenter, open-label, randomized clinical trial including 14 secondary and academic hospitals in the Netherlands. In the Netherlands, a secondary hospital is a non-academic hospital where care is provided by medical specialists. Enrollment began on 8 April 2020. Eligible patients were at least 18 years, admitted to the hospital for COVID-19 proven by a SARS-CoV-2 genome detectable in a RT-PCR test in the previous 96 h. Patients with documented IgA deficiency or on mechanical ventilation for >96 h at the time of screening were excluded. Concurrent inclusion in another interventional study aimed at COVID-19 treatment was prohibited. Upon the discretion of the research physician, eligible patients identified through screening were not included when care had entered a terminal phase or a patient had already improved significantly to a fit for discharge level.

Recovered COVID-19 patients who could potentially participate as plasma donors were informed on this option by social media notifications. Interested donors could apply by email. Eligible donors had RT-PCR-confirmed SARS-CoV-2 infection and were asymptomatic for minimally 14 days. Written informed consent was obtained and a questionnaire was sent by email via Gemstracker. ConvP donors were recruited and screened by Sanquin Blood Supply (Dutch blood bank) according to existing guidelines (Appendix 1 study protocol, section 8.3 on donor eligibility criteria). Donors could voluntarily donate up to maximum of four times at 1-week intervals. A single serum tube for SARS-CoV-2 antibody assessment was drawn on the first day of donation. Only donor ConvP with SARS-CoV-2-neutralizing antibodies confirmed by ELISA and having a SARS-CoV-2 PRNT and a PRNT50 titer of minimally 80 was used[37,42]. For each patient, we selected the plasma with the highest PRNT50 titer from the donor pool available at the time of inclusion. Donors completed a detailed questionnaire on their medical history and COVID-19 clinical symptoms.

**Study procedures and endpoints**. Participants provided written informed consent, had blood group determined, and were subsequently randomly assigned via a web-based system ALEA at a 1 : 1 ratio to the current SoC with or without the addition of 300 mL ConvP including SARS-CoV-2-neutralizing antibodies with a known adequate PRNT50. The chosen volume reflects the standard volume of one plasma unit produced by Sanquin Blood Supply and was comparable to the volume (280 mL) of ConvP used in studies for SARS-CoV[11]. ConvP was administered intravenously on the day of inclusion. Patients without a clinical response and a persistently positive RT-PCR could receive a second unit of ConvP after 5 days. Off-label use of European Medicines Agency (EMA)-approved drugs as a treatment for COVID-19 was allowed in hospitals where this was part of the SoC. We scored the clinical status with the ordinal eight-point WHO COVID-19 disease severity scale on days 1, 15, and 30[54]. Serum samples and nasopharyngeal swabs were collected at inclusion preceding treatment and on day 3, 7, and 14. One serum tube per participant for immunological and virological assays were collected at enrollment and on day 7 and 14. This material was used for the detection of antibodies by ELISA and, with sufficient serum available, PRNT. The primary endpoint of the study was overall mortality until discharge from hospital or a maximum of 60 days after admission, whichever came first. Key secondary endpoints were the improvement on the eight-point WHO COVID-19 disease severity scale on day 15 and day 30, hospital length of stay, SARS-CoV-2 shedding from the airways, impact of ConvP on humoral immunity, and inflammation. Safety of ConvP was recorded as any plasma-related transfusion reaction or death.

**Clinical data**. All principal investigators and the sites' study teams were trained before any study procedure through site initiation meetings. Baseline characteristics and medical history were recorded in the electronic case record (eCRF) formed by a trained research physician. The comorbidities were assigned using the following definitions; hypertension was defined as hypertension reported in the medical history, including hypertension with or without end organ damage, and also both hypertension for which medication was given and for which no medication was given. Diabetes mellitus was defined as either type 1 or type 2. This also included diabetes with or without end organ damage. Cardiac history was defined

as any chronic disorder of the cardiac function that made the subject eligible for yearly influenza vaccination according to the Dutch guidelines[55]. A history of pulmonary disease was defined as any chronic pulmonary condition, which required inhalators or systemic medication, or follow-up with a pulmonologist. A history of cancer was defined as any active cancer in the previous 5 years (cutaneous basal cell carcinoma was not included). A history of immunodeficiency was defined as any documented clinical relevant immunocompromised condition or active use of immunosuppressants. A history of chronic kidney disease was defined as any kidney disorder due to an estimated glomerular filtration rate below 60 ml/min, macroalbuminuria, peritoneal or hemo-dialysis, or prior kidney transplantation. A history of liver cirrhosis was defined as liver cirrhosis classified as Child-Pugh A or higher. Admitted participants were assessed as inpatients by the research physicians at the study time points on clinical endpoints. The use of experimental medication for COVID-19 was recorded in the eCRF for (hydroxy)chloroquine, lopinavir/ritonavir, and remdesivir. Discharged patients were contacted at the study time points to assess the clinical status. The most recent routinely measured serum biomarkers for COVID-19 severity were collected in the eCRF and were generally available for CRP (1 missing), LDH (3 missing), bilirubin (7 missing), lymphocyte count (8 missing), and ferritin (29 missing). All sites collected and entered data into an eCRF (OpenClinica). Independent data monitors scrutinized the data quality and solved inconsistencies in the eCRF. A.G., C.C.E.J., B.J.A.R., C.R., and G.R.v.P. extracted and analyzed the data.

**SARS-CoV-2 plaque-reduction neutralization test**. We analyzed serum samples of donors and patients for the presence of neutralizing antibodies by performing a PRNT with the SARS-CoV-2 virus (German isolate; GISAID ID EPI_ISL 406862; European Virus Archive Global #026V-03883)[37]. We 2-fold serially diluted heat-inactivated samples and added 400 plaque-forming units to each well, then incubated at 37 °C for 1 h before placing the mixtures on Vero-E6 cells. After 8 h of incubation, we fixed and stained the cells and counted the number of infected cells per well by using an ImmunoSpot Image Analyzer (CTL Europe GmbH, https://www.immunospot.eu). The serum neutralization titer is the reciprocal of the highest dilution resulting in an infection reduction of >50% (PRNT50). We considered a titer ≥20 to be positive.

**SARS-CoV-2 antibody ELISA assays**. Serum was tested for the presence of SARS-CoV-2 total Ig and IgM antibodies against RBD in the Wantai ELISA test (Wantai Biological, Beijing). We previously showed that a positive total Ig or a IgM with an OD ratio > 10 (which equals an OD of 2.0), correlates closely with PRNT50 of at least 80[36]. SARS-CoV-2 virus nucleocapsid protein (N-protein)-specific antibodies in serum were measured by ELISA using COVID-19 IgG ELISA (Tecan, 30177447) and COVID-19 lgM ELISA (Tecan, 30177448) according to the manufacturer's instructions. Positive cutoff for these ELISAs was 11 units.

**Serum cytokine measurements**. Cytokines IL-6, TNFα, IFNγ, IL-1β, IL-2, IL-4, IL-10, and IL-12p70 in serum of COVID-19 patients and plasma donors were measured using Simple Plex Cytokine Screening Panel cartridges (SPCKE-PS-003426, Bio-Techne) with the Ella Next Generation ELISA system (Bio-Techne). The lower and upper limit of quantification was 0.21 and 840 pg/ml for IL-1β, 0.64 and 990 pg/ml for IL-2, 0.32 and 1290 pg/ml for IL-4, 0.28 and 2652 pg/ml for IL-6, 0.46 and 5530 pg/ml for IL-10, 0.46 and 2.7 pg/ml for IL-12p70, 0.17 and 4000 pg/ml for IFNγ, and 0.3 and 1160 pg/ml for TNFα.

**Real-time PCR detection**. Real-time PCR detection of SARS-CoV-2 was performed using the SARS-CoV-2 test on a cobas® 6800 system (Roche Diagnostics)

**Viral culture**. Vero cells, clone 118, were used for isolation of infectious SARS-CoV-2 from respiratory tract samples. Samples were cultured for 7 days and, once CPE was visible, the presence of SARS-CoV-2 was confirmed with immuno-fluorescent detection of nucleocapsid proteins.

**Sample size and statistical analysis plan**. Baseline descriptive statistics are provided as median with IQR or mean with 95% CI for continuous variables and as count with percentage for categorical variables. A Mann–Whitney $U$-test, a $t$-test, or a $\chi^2$-test was used to describe differences in these baseline statistics. With an anticipated 50% overall mortality reduction from 20% (as the reported mortality in hospitalized patients in the Netherlands when the protocol was designed) and with a control to intervention ratio of 1 : 1, 426 patients were needed for the study to have 80% power with a global α of 0.05 and adjusted α for the primary endpoint of 0.0480, accounting for one interim analysis. Due to the premature interruption of the trial and resulting in lower event rates, we present both the results of the multivariable (adjusted) logistic regression analysis as originally planned as the principle analysis, as well as the unadjusted univariable analysis. The effect of plasma therapy on overall mortality was estimated by logistic regression models adjusted for the independent factors at inclusion sex, age, ICU admission, CRP, absolute lymphocyte count, bilirubin, and FiO2. A two-sided Wald's test on the OR with 95% CI of the treatment effect based on the multivariable model was planned to assess whether ConvP reduces mortality at the adjusted α-level of 0.0480. A

proportional odds ordinal logistic regression model was used to estimate the odds of being worse on the eight-point WHO COVID-19 disease severity scale at day 15 and day 30 after inclusion, and adjusted for the seven factors mentioned above. This model was used to test the hypothesis that the treatment to control OR is equal to 1. The impact of ConvP therapy on the length of hospital stay was analyzed both with a proportional hazards model for the subdistribution of hospital discharge as proposed by Fine and Gray (1999)[56] and by reporting the cause-specific hazards using Kaplan–Meier plots. The correlation between time and PRNT50 were assessed by Spearman's correlation coefficients. The viral load (copies/ml) was analyzed using a mixed-effects model with random intercepts and random slopes. A linear effect of time was used in the model. An interaction effect between time and treatment was included in the model, to allow for different evolution over time between the treatment arms and to assess whether the rate of decrease is different between the two arms. The outcome was transformed using the logarithmic function to avoid deviations from the normality and homoscedasticity assumptions of the model. The value of 0.001 was added to zero values of viral load before applying the transformation to avoid minus infinity values.

**Ethical considerations**. The study was reviewed and approved by the institutional review board of the Erasmus University Medical Center. Written informed consent was obtained from every patient or legal representative. The DSMB consisted of a professor in biostatistics, an infectious diseases specialist, and an intensivist. They reviewed the safety of the participants on a regular basis and recommended the study team regarding the further conduct of the study at predefined time points. Findings are reported according to the CONSORT (Consolidated Standards of Reporting Trials) statement. The study was registered as NCT04342182 at clinicaltrials.gov.

**Trial registration**. Clinicaltrials.gov:NCT04342182.

**Reporting summary**. Further information on research design is available in the Nature Research Reporting Summary linked to this article.

## Data availability
The datasets generated and/or analyzed during the current study are available from the corresponding author on request. Source data are provided with this paper.

## Code availability
The codes generated during the current study are provided with this paper.

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

## Acknowledgements

We thank all plasma donors who volunteered in great numbers and all patients who participated in the study; all colleague internist-infectiologists for giving the study team at Erasmus MC the time to execute this study during unprecedented times; Professor Stephanie Klein-Nagelvoort Schuit, Professor Annelies Verbon, and Professor Charles Boucher, who sadly passed away on 26 February, 2021 for their support; the institutional review board at Erasmus MC for their efficient review of the application and several amendments on short notice; HOVON, in particular Monique Steijaert and Henk Hofwegen, for helping with IRB-related procedures and for programming the ALEA online randomization tool; Aldert Lamore, for programming the eCRF; and Sanquin Blood Supply Netherlands for collecting ConvP from hundreds of donors. We thank the members of the data safety monitoring board Dr. JL Nouwen, Professor H Boersma, and Dr. B Van der Hoven; the Erasmus foundation for financial support, and all those who donated to the foundation for this study, in particular Eduard Haegens and Marleen Hoex from Ypsilon, who made it financially possible to start the trial; Health Holland for the LSHM20056 grant to perform immunological assays; the department of virosciences and the department of immunology (in particular Rik Ruijten, Inge Brouwers - Haspels, Caoimhe Van der Wel – Kiernan, Harm de Wit, Marjan van Meurs, Manzhi Zhao, Melisa Castro Eiro, Merel Wilmsen, Tamara van Wees, and Tessa Alofs), and all laboratory technicians and medical students (in particular Renée Deckers, Freya Huijsmans, Niek van der Maas, Miliaan Zeelenberg, Zgjim Osmani, Jari Hofmans, and Lieke Heijnen), and Frank van Vliet for helping with real-time data and sample collections throughout the country; and BMW, the Netherlands, for providing two cars free of charge for sample collection. We also thank Dr. Corine Delsing from Medisch Spectrum Twente, Dr. Jiri Wagenaar from Noordwest Ziekenhuisgroep, and Robin Soetekouw from Spaarne Gasthuis for engaging as a study site, all study nurses and trial coordinators in the participating centers, and in particular Siepke Hiddema, Frances Greven, and resident Sander Albers, and many others that volunteered their time and effort for this study.

## Author contributions

A.G., C.C.E.J., C.R., P.t.B., G.P., M.P.G.K., and B.J.A.R. designed the study. G.P. is the study statistician. A.G., C.C.E.J., and B.J.A.R. designed the database and eCRF. J.v.K., N.M.A.O., M.P.G.K., B.L.H., and C.G. were responsible for the virological assays, RT-qPCR, PRNT, and the anti-SARS-CoV-2 total Ig and IgM against RBD. Y.M., M.S., P.D.K., and T.S. were responsible for the immunological analyses. F.H.S. organized plasma donations at the Dutch blood Bank (Sanquin Blood Supply). All other authors recruited patients and collected study data. A.G., C.C.E.J., C.R., G.P., and B.J.A.R. analyzed the data. B.J.A.R. and C.R. wrote the first draft of the paper and all authors reviewed the paper.

## Competing interests

The Netherlands Organisation for Health Research and Development (ZonMW, grant agreement 10150062010008, B.L.H.), the Erasmus foundation grant (B.J.A.R.), and the PPP Allowance (grant number LSHM20056, P.D.K.) made available by Health~Holland, Top Sector Life Sciences & Health, to stimulate public-private partnerships. The remaining authors declare no competing interests.

## Additional information

**Peer review Information** *Nature Communications* thanks the anonymous reviewer(s) for their contribution to the peer review of this work. Peer reviewer reports are available.

