## [Peer Review File · Nature Communications]

Reviewers' Comments:

Reviewer #1:

Remarks to the Author:

General comments

Gharbharan et al. have conducted a highly important study on the impact of immune plasma administered to COVID-19 patients. The study is performed with completely satisfactory methodological rigor in all regards, within the limits set by the clinical-ethical situation. The authors found no evidence of beneficial effects, as measured by clinical observation or virological and immunological laboratory tests, including viral load in pulmonary secretions. The implication, rather than a direct observation, of their results is that earlier treatment might be beneficial. The current study, although interrupted in a controlled form, strengthens the evidence from a previous one of lack of effects, although the neutralizing titers of the current donor plasmas were one average more than ten-fold higher.

It is possible that higher neutralizing capacity and therefore more beneficial effects might also be obtained with higher titer plasma (covered in lines 325 forwards) or cocktails of neutralizing monoclonal antibodies (mentioned on line 348). Future studies testing the refinement of the hypotheses in these regards that the current study has helped sharpen will be equally important.

The authors are encouraged to expand explanations and discussion of why no stratification or correlation within the current data was possible to test whether indeed the highest-neutralizing titer donor plasma in the lowest-neutralizing titer recipients had any detectable beneficial effects that would have been blunted in combinations with lower donor-to-recipient ratios. Is this merely a matter of too few data points?

Because of considerable confusion in the rapidly growing COVID-19 literature, it would be valuable to include a paragraph with a reference to a general review of neutralization of virus infectivity by antibodies, which covers mechanisms and explains which anti-viral effects do not count as neutralization.

Specific comments

62: A very reasonable conclusion: "Together, these data indicate that the variable effectivity observed in trials on convalescent plasma for COVID-19 may be explained by the timing of treatment and varying levels of preexisting anti-SARS-CoV-2 immunity in patients. It also substantiates that convalescent plasma should be studied as early as possible in the disease course or at least preceding the start of an autologous humoral response." – except that "convalescent" is not the right term here. These plasmas should be collected as early as possible in the disease course, as the authors indeed indicate, well before convalescence starts.

216 et pass., forwards: <1:320, >1:2560 etc., this is a common way of writing titers, which nevertheless remains formally incorrect. Since the : is the old-fashioned sign for fractions (/), the ratio 1/320 is, of course greater than 1/2560, whereas, obviously, the authors use the > and < signs with the opposite meaning. 1/(<320) etc would be correct. To avoid the latter, admittedly awkward designations, the authors can make an early blanket statement that reciprocal titers = dilution factors are given, <320, >2560.

319: this is a complex relationship in that at one stage of disease neutralizing titers are highest in the most severely ill COVID-19 patients, then all but disappear in moribund cases, at least in some studies, similarly in the earlier epidemic with the SARS-CoV(-1) virus.

Reviewer #2:

Remarks to the Author:

Gharbharan et al report their multicenter open-label randomized clinical trial investigating the use of convalescent plasma from donors with high titers of neutralizing antibodies. No overall clinical benefit was found in the 86 patients enrolled before the study was terminated early because the vast majority of the patients already had potent neutralizing anti-SARS-CoV2 antibodies at the time of hospital admission. The authors evaluate the virological and immunological responses of these patients.

The manuscript is well written, but the overall number of patients in the trial is quite low. The addition of information on the plasma donors, serum from 37 RT-PCR-confirmed COVID-19 patients in the month before the start of the study, and healthy patients adds to the complexity and makes the manuscript somewhat less straight forward and harder to follow. Much of the laboratory data presented on the donors and patients are already well known at this point.

Discussion:

Page 10, please comment on why the donors in ConCOVID study would have 10 times higher titer than the Placid trial. Is this related to the location, the selected donor population, or assay used?

Please mention the role of a higher dose of convalescent plasma, since all but 2 patients in ConCOVID received only 1 unit. The average dose for a non-COVID plasma transfusion is around 3 units. Please explain why more patients didn't get a second dose.

Minor: when describing antibodies in the manuscript, use of both anti- and antibodies is redundant, for example anti-SARS-CoV-2 antibodies. This should be corrected throughout the manuscript.

Page 14, line 386, please define the term "secondary hospital"

We thank the reviewers of Nature Communications for the very hard work in these unprecedented research arena and for their time in reviewing our work. Below are the replies to the questions that were raised.

Reviewer 1:

This reviewer writes *“the authors are encouraged to expand explanations and discussion of why no stratification or correlation within the current data was possible to test whether indeed the highest-neutralizing titer donor plasma in the lowest-neutralizing titer recipients had any detectable beneficial effects that would have been blunted in combinations with lower donor-to-recipient ratios. Is this merely a matter of too few data points?”*

Author’s answer:

It is indeed impossible to evaluate the correlation between titers and clinical benefit given the fact that only half of the 86 patients included received convalescent plasma. Furthermore, the rigorous selection criteria regarding plasma donors that we used resulted in a median PRNT50 titer of 1/640 which is around the top 10% of titers in donors. So the granularity of the data as well as the sample size do not allow to further analyse the titer-response correlation.

This reviewer states *“because of considerable confusion in the rapidly growing COVID-19 literature, it would be valuable to include a paragraph with a reference to a general review of neutralization of virus infectivity by antibodies, which covers mechanisms and explains which anti-viral effects do not count as neutralization”*.

Author’s answer:

We also found it an asset for this paper to include a paragraph in the introduction explaining the mechanism of viral neutralization. We therefore added this paragraph in the introduction of the paper *“Neutralizing antibodies can cause a reduction of virus infectivity by binding to the surface of viral particles which results in blocking one of the steps of the viral replication cycle, also known as virus neutralization.(7) Neutralizing SARS-CoV-2 antibodies recognize regions of the Spike protein, mainly the receptor binding domain (RBD), and inhibit viral infectivity by several mechanisms. The most important one is blocking the RBD-ACE-2 receptor interaction and a such preventing the attachment of SARS-CoV-2 to the epithelial cell surface”*

Postadres

Postbus 2040
3000 CA Rotterdam

Bezoekadres

Dr. Molewaterplein 40
3015 GD Rotterdam

Parkeergarage

Dr. Molewaterplein 40
3015 GD Rotterdam

Head of Department

Sectie Infectieziekten

Mw. Prof. dr. A. Verbon

Internists-Infectiologists

Dr. J.L. Nouwen
Dr. B.J.A. Rijnders
Mw. Dr. .C.A.M. Schurink
Mw. Dr. T.E.M.S. de Vries
Mw. Dr. H.I. Bax
Dr. C Rokx
Mw. Dr. E van Nood
Mw. Drs. M de Melo Mendonca
Dr. L Slobbe

Specific comments

The reviewer concludes that our study draws a *very reasonable conclusion*: “Together, these data indicate that the variable effectivity observed in trials on convalescent plasma for COVID-19 may be explained by the timing of treatment and varying levels of preexisting anti-SARS-CoV-2 immunity in patients. It also substantiates that convalescent plasma should be studied as early as possible in the disease course or at least preceding the start of an autologous humoral response.” – except that “convalescent” is not the right term here. These plasmas should be collected as early as possible in the disease course, as the authors indeed indicate, well before convalescence starts. (line 62)

Author’s answer:

We believe a mix up has taken place. In this line we wanted to suggest that the administration of convalescent plasma should be investigated as early as possible in the disease course. We rewrote this sentence in the abstract.

The reviewer notes that: $<1:320, >1:2560$ etc., *this is a common way of writing titers, which nevertheless remains formally incorrect. Since the : is the old-fashioned sign for fractions (/), the ratio 1/320 is, of course greater than 1/2560, whereas, obviously, the authors use the > and < signs with the opposite meaning. 1/(<320) etc would be correct. To avoid the latter, admittedly awkward designations, the authors can make an early blanket statement that reciprocal titers = dilution factors are given, $<320, >2560$. (line 216 et pass.)*

Author’s answer:

We changed the way the neutralizing titers are reported into diluting factors.

The reviewer commented *this is a complex relationship in that at one stage of disease neutralizing titers are highest in the most severely ill COVID-19 patients, then all but disappear in moribund cases, at least in some studies, similarly in the earlier epidemic with the SARS-CoV(-1) virus. (line 319)*

Authors answer:

This seems to be just a comment but not a request for a change or additional data. We made no changes in regard to this comment. Also, no moribund cases were included in this study.

Reviewer 2:

This reviewer stated that *the manuscript is well written, but the overall number of patients in the trial is quite low.*

Author’s answer:

We agree that the overall number is quite low which justifies some precautions with regard to conclusions on hard clinical endpoints. We already discuss this in detail in the discussion.

The reviewer also stated *the addition of information on the plasma donors, serum from 37 RT-PCR-confirmed COVID-19 patients in the month before the start of the study, and healthy patients adds to the complexity and makes the manuscript somewhat less straight forward and harder to follow.*

Author’s answer:

Given the relatively small sample size in the main study, we wanted to confirm our conclusions in an

independent but comparable patient group. Therefore, we really think that these additional data are required.

The reviewer stated that *much of the laboratory data presented on the donors and patients are already well known at this point.*

Author's answer:

Part of the data in our paper were published on the preprint server already early July 2020. At that time several of the results we described were reported for the very first time in our paper e.g. the very number of antibody positive patients at baseline, the lack of a difference in (neutralizing) antibodies on day 7 after randomization. This preprint paper was broadly covered in the scientific media (and other media) and the paper was downloaded >13.000 times and the abstract read almost 50.000 times. This is illustrated by the fact that the study has been extensively cited in published papers and is cited >100 times on google scholar. Therefore, we agree that the data are now well-known but this is partially the result of this study and its preprint publication. The Nature communications version of the paper has additional data (e.g on viral kinetics, inflammation markers etc.) that are not yet broadly reported.

Discussion

The reviewer asks to elaborate on *"why the donors in ConCOVID study would have 10 times higher titer than the Placid trial. Is this related to the location, the selected donor population, or assay used? (page 10)*

Author's answer:

In the PLACID trial no real-time neutralizing assays were performed to determine the titers prior to the transfusion. These were done in retrospect. We however, did determine the PRNT50 real-time. We therefore could select to only administer the plasma units with the highest available PRNT50.

The reviewer asks to *"mention the role of a higher dose of convalescent plasma, since all but 2 patients in ConCOVID received only 1 unit". The average dose for a non-COVID plasma transfusion is around 3 units. Please explain why more patients didn't get a second dose.*

Author's answer:

The use of plasma for other diseases is rarely related to administration of virus neutralizing antibodies, so comparing doses across different disease is a bit like apples and oranges. That being said, we agree that no formal dose finding studies have ever been performed regarding virus neutralizing activity of 1 versus more plasma units and it is possible that we underdosed with one unit of plasma. However, the protocol allowed for a second transfusion. We are aware of a large RCT in Belgium that is now fully enrolled (460 patients) in which 4 units of plasma are used (NCT4429854). So future will tell if we underdosed or not.

The reviewer asks *"when describing antibodies in the manuscript, use of both anti- and antibodies is redundant, for example anti-SARS-CoV-2 antibodies. This should be corrected throughout the manuscript"*

Author's answer:

We corrected the redundancy mistake of anti-SARS-CoV-2 antibodies in the manuscript.

The reviewer commented *define the term "secondary hospital"*. (page 14, line 386)

Author's answer:

We added a sentence to explain secondary hospitals in the Netherlands.

Reviewers' Comments:

Reviewer #1:

Remarks to the Author:

The authors have comprehensively and conscientiously responded to all points raised in the reviews.

Regarding:

"The reviewer commented this is a complex relationship in that at one stage of disease neutralizing titers are highest in the most severely ill COVID-19 patients, then all but disappear in moribund cases, at least in some studies, similarly in the earlier epidemic with the SARS-CoV(-1) virus. (line 319)

Authors answer:

This seems to be just a comment but not a request for a change or additional data. We made no changes in regard to this comment. Also, no moribund cases were included in this study."

- the suggested change was that if the authors agree that the literature suggests an opposite relationship between antibody titers and severity of disease in the most severely ill cases, then a small addition to this passage would improve the precision of the account of the literature.

Reviewer #3:

Remarks to the Author:

The study by Gharbharan et al. reports the results of a study of convalescent plasma for severe COVID-19 in the Netherlands during the early days of the pandemic. This study was stopped when the investigators determined that it made no sense to continue because most of their patients already had endogenous antibody. Many, including this reviewer, feel that this was an unfortunate decision since this was a very well-designed trial and the available results at the time the trial was stopped revealed a large difference in mortality between treated and standard of care group. Although the authors state that convalescent plasma 'did not improve survival' the fact of the matter is that the mortality in the plasma group was 14% while that in the standard of care was 26%. I pose the following question to the investigators – if you were a patient, which group would you prefer to be in – plasma or SoC? I think most people would choose the plasma group. I make this point because the paper is written with assumption that failure to find a statistical difference means there is no difference, and the wording should be modified to better reflect reality. The decision to stop the trial based on the detection of antibodies in the patient cohort is perplexing when one considers that antibodies made in the convalescent period are often much more effective at viral neutralization through somatic mutation than early responses. Hence, one is left to wonder what the outcome of this work would have been had the trial proceeded to conclusion. Another peculiarity of this study relative to other clinical trials is that the authors did not see a viral clearance effect in the plasma group. Given that various studies have reported a viral reducing effect, the absence of such a finding in this study is unexplained. The early termination of this trial significantly weakened the study and the conclusions of the work.